# Variations in Human Milk Metabolites After Gestational Diabetes: Associations with Infant Growth

**DOI:** 10.3390/nu17091466

**Published:** 2025-04-26

**Authors:** Alice Fradet, Line Berthiaume, Laurie-Anne Laroche, Camille Dugas, Julie Perron, Alain Doyen, Étienne Audet-Walsh, Julie Robitaille

**Affiliations:** 1Centre NUTRISS—Nutrition, Health, and Society, Institute of Nutrition and Functional Foods (INAF), Université Laval, Quebec City, QC G1V 0A6, Canada; alice.fradet.1@ulaval.ca (A.F.); cam_dugas@hotmail.com (C.D.); julie.perron@fsaa.ulaval.ca (J.P.); 2Endocrinology—Nephrology Research Axis, CHU de Québec Research Centre, Université Laval, Quebec City, QC G1V 4G2, Canada; line.berthiaume@crchudequebec.ulaval.ca (L.B.); etienne.audet-walsh@crchudequebec.ulaval.ca (É.A.-W.); 3Centre de Recherche en Reproduction, Développement et Santé Intergénérationnelle (CRDSI), Quebec City, QC G1V 0A6, Canada; 4School of Nutrition, Faculty of Agricultural and Food Sciences, Université Laval, Quebec City, QC G1V 0A6, Canada; laurie-anne.laroche.1@ulaval.ca; 5Institute of Nutrition and Functional Foods (INAF), Université Laval, Quebec City, QC G1V 0A6, Canada; alain.doyen@fsaa.ulaval.ca; 6Department of Food Sciences, Faculty of Agricultural and Food Sciences, Université Laval, Quebec City, QC G1V 0A6, Canada; 7Department of Molecular Medicine, Faculty of Medicine, Université Laval, Quebec City, QC G1V 0A6, Canada

**Keywords:** gestational diabetes mellitus, human milk, metabolite, infant growth

## Abstract

Background/Objectives: Gestational diabetes mellitus (GDM) is a condition characterized by hyperglycemia and is associated with increased risk of obesity and diabetes in exposed children. Differences in human milk composition between women with (GDM+) and without GDM (GDM-) suggest that GDM could impact milk production and composition, potentially influencing infant growth. However, this association remains poorly understood. The objective was to study the association between GDM and human milk composition and its influence on infant growth, focusing on metabolites and bioactive molecules involved in energy metabolism. Methods: Using a cross-sectional design, 24 metabolites were measured by GC-MS in human milk obtained at 2 months postpartum from 20 GDM+ women and 29 GDM- women. Anthropometric measures, as well as lipid and glycemic profiles, were collected. Infant weight and length data were obtained from health records. Results: Human milk metabolites significantly differ between GDM+ and GDM- mothers, with higher levels of myristic acid, glycerol, uracil, arachidonic acid, and cholesterol in GDM+ milk (*p* < 0.05). Specific human milk metabolites showed distinct correlations with maternal glycemic as well as infant growth, depending on GDM status. While maternal glycemia was associated with succinate and malate in all groups, maternal glycemia was specifically correlated with valine and glutamate in GDM+ mothers. Additionally, in GDM+ women, α-ketoglutarate and glycine were negatively correlated with infant growth. Conclusions: The results of this study suggest that GDM can influence the mother’s health beyond delivery, impacting the mammary gland biology with effects on the human milk composition. Further, correlations with infant growth suggest that GDM-dependent variations in milk composition potentially influence infant growth and metabolism.

## 1. Introduction

Gestational diabetes mellitus (GDM) is characterized by hyperglycemia that develops during pregnancy and usually disappears after delivery [1]. It is the most common complication among pregnant women [2]. Beyond short-term complications, such as increased risks of fetal macrosomia, preeclampsia, and cesarean delivery, GDM can affect both the mother and the child [1]. Indeed, it has been estimated that the risk for women with GDM (GDM+) of developing type 2 diabetes within 5 years after pregnancy is 50% higher [3], and the risk of developing cardiovascular diseases is two to three times higher [4]. Moreover, children exposed in utero to GDM have a higher risk of developing obesity, type 2 diabetes, and, for females, GDM [5]. The identification and understanding of key mechanisms associated with this increased risk is of paramount importance.

Despite being an intensive research area, there is still an incomplete understanding of the mechanisms driving GDM, as well as its downstream impacts, such as on the mother, following delivery. For instance, GDM occurs during pregnancy while the mammary gland is preparing for lactation/breastfeeding, and it is yet unclear how GDM modulates the mother’s capacity to produce milk [6]. Indeed, human milk production is a complex process mediated by the mammary gland, a dynamic organ that develops primarily after birth [7]. During pregnancy, there is a release of a reproductive hormonal cocktail, such as estrogen, progesterone, and prolactin, that induces the proliferation and maturation of mammary epithelial cells. These cells form alveolar progenitor cells, which then differentiate to acquire the capacity to synthesize and secrete milk [8]. Hormones play key roles in this process. They promote the development of mammary alveoli, increase transcription of milk component genes, stimulate the closure of epithelial cell tight junctions during early lactation, and mediate milk synthesis in response to infant suckling [9]. In the context of GDM, alterations in maternal metabolic and hormonal environments may influence the levels or actions of these hormones, potentially impacting mammary function and milk composition [10]. In addition, metabolic hormones such as insulin participate in the formation and secretory differentiation of alveoli, facilitating the transition of the mammary gland to a functional milk secretory organ after birth [11]. Breastfeeding is a very complex and costly mechanism for the mother’s body, and it lasts for several months [12,13,14].

Human milk contains a variety of bioactive metabolites such as macronutrients, including proteins (especially casein, lactalbumin, and lactoferrin), lipids (triglycerides, phospholipids), and carbohydrates (mainly human milk oligosaccharides) [15]. In addition, human milk contains a diversity of micronutrients, including amino acids such as alanine, glycine, leucine, and glutamate, which play a crucial role in child nutrition and development [16]. It also contains various molecules related to energy metabolism connected to glycolysis and the tricarboxylic acid (TCA) cycle, such as lactate, α-ketoglutarate, succinate, citrate, and malate [17]. The TCA cycle is a key element in maintaining energy production, and the intermediates of this cycle are essential metabolites in cellular development and growth [18]. Since GDM represents a dysregulation of energy metabolism, the study of how potential variations of the TCA cycle in human milk influence the infant’s development might provide valuable information to better understand the impact of GDM.

The composition of human milk is not universal, and maternal conditions such as obesity or GDM may influence its composition and the infant’s energy metabolism [19]. In support of this hypothesis, we have previously shown differences in triglycerides in the milk of women with prior GDM (GDM+) compared to women without (GDM-) [20]. Additionally, differences have been observed in the endocannabinoidome system, particularly in metabolites related to feeding signals, which correlated with infant growth [21]. More specifically, we have previously shown that circulating N-palmitoylethanolamine (PEA) and 2-arachidonylglycerol (2-AG) were higher in the human milk of GDM+ women than in GDM- women [21]. The levels of non-omega-3 N-acyl-ethanolamines (NAEs) in human milk were negatively correlated with weight-for-age z-score among GDM+ offspring [21]. Different studies, including those from our research group, have shown that GDM+ human milk differs from that of GDM- human milk [20,21,22,23,24]. However, few have focused on the metabolic profile of milk, particularly metabolites and bioactive molecules related to energy metabolism, and how these may influence infant growth trajectories. Currently, there is insufficient data on how GDM influences human milk composition and its impact on infant health.

Thus, the objective of this study was to investigate the metabolic impact of GDM on human milk composition. We notably focused on amino acids and energy metabolism-related molecules, examining how their levels correlate with infant growth. We also aimed to explore how the maternal metabolic profile (anthropometric, glycemic, and lipid profiles) is associated with human milk components. The underlying hypothesis is that GDM modulates the mammary gland maturation required for breastfeeding. Thus, after delivery, GDM leads to long-term metabolic reprogramming of the mammary gland that affects the milk composition and, ultimately, influences infant growth. For these analyses, we conducted metabolomics on samples from a well-established cohort [20,21] that contained milk from both GDM- and GDM+ women.

## 2. Materials and Methods

### 2.1. Study Design and Participants

This cross-sectional study used data from a cohort of 32 GDM+ women and 30 GDM- women and their infants [20,21]. Baseline data and samples from GDM+ women, who took part in an 18-month intervention study starting 2 months postpartum, were included for this cross-sectional study. GDM+ women under prenatal care by clinicians at the two main hospitals with neonatal care units in Quebec City, Canada, were invited to join this clinical trial (NCT02872402) 2 months after delivery. Additionally, recruitment emails were sent to the Université Laval community. The recruitment of 32 GDM+ women occurred between January 2017 and September 2019. The 30 GDM- women were recruited via emails from the Université Laval community between March and September 2020 (NCT04263675). These participants attended a single visit at the Institute of Nutrition and Functional Foods (INAF) in Quebec, Canada. To be included, GDM+ and GDM- participants needed to meet the following criteria: fluent in French, a singleton and term pregnancy (>37 weeks), at least 18 years old, and a pre-pregnancy BMI of 18.5 kg/m^2^ or higher. Exclusion criteria were multiple pregnancies, preterm delivery (<37 weeks), a history of bariatric surgery, or planning a pregnancy in the next year. For GDM- women, an additional inclusion criterion was that they must not have had a GDM diagnosis in a previous pregnancy.

Only mother–infant pairs who had complete data and human milk samples at 2 months postpartum were included in the current analysis, 29 GDM- and 20 GDM+ women. This study adhered to the guidelines of the Declaration of Helsinki and received approval from the Centre Hospitalier Universitaire de Québec Ethics Committee (2017-3225 and 2020-5075).

### 2.2. Maternal and Infant Data

During a visit at 2 months postpartum, maternal weight was recorded using a calibrated scale to the nearest 0.1 kg, and height was measured with a stadiometer, from which BMI (kg/m²) was calculated. Body composition and fat distribution were assessed using a dual energy X-ray absorptiometry scanner (DXA) [25]. Fasting blood samples were collected, followed by a 75 g 2 h oral glucose tolerance test. Plasma glucose levels at 0 and 2 h were measured enzymatically, and insulin levels at 0 and 2 h were determined using the ADVIA Centaur CP Insulin (IRI) assay [26,27]. This in vitro diagnostic immunoassay employs direct chemiluminescent technology to quantify insulin in serum. The intra-assay CVs for low, intermediate, and high insulin concentrations were 4.6%, 3.2%, and 3.3%, respectively, while the inter-assay CVs were 5.9%, 2.6%, and 4.8%. The homeostasis model assessment for insulin resistance (HOMA-IR) index was calculated as follows: [fasting insulinemia (lU/L)·fasting glycemia (mmol/L)/22.5] [28]. The area under the curve (AUC) for glucose during the oral glucose tolerance test (OGTT) was calculated using the trapezoidal rule based on glucose values measured at fasting, 1 h, and 2 h. Serum total cholesterol, triglyceride, and HDL cholesterol concentrations were measured using a Roche/Hitachi Modular system (Roche Diagnostics) [29], and serum LDL cholesterol levels were calculated with the Friedewald equation [30].

Infant weight and length data were obtained from health records which included measured values of weight and length by healthcare providers. Weight-for-age (WAZ), weight-for-length (WLZ), and length-for-age (LAZ) sex-specific z-scores at birth and at 2 months were calculated using the World Health Organization’s growth standard charts [31]. Growth between birth and 2 months, expressed as delta (Δ) values, was also calculated.

### 2.3. Human Milk Collection and Processing

The women were provided written instructions to collect 30 to 60 mL of human milk at the end of a feeding session. They recorded the date and time of collection and stored the milk in sterile cups in their home freezers. The frozen samples were then transported to the research center in insulated bags containing ice packs. All human milk samples were then stored at −80 °C. The frozen samples were defrosted at 0 °C to 4 °C, and the whole human milk samples were vortexed at high speed for 30 s to ensure sample homogeneity immediately before aliquoting and storing at −80 °C until batch analysis. The composition of the milk, including its energy, triglyceride, lactose, and protein content, had been previously reported [20]. The research assistants who processed the milk samples were blinded to the participants’ group assignment.

The characteristics of the participants have already been presented [20,21].

### 2.4. Metabolites in Human Milk

Ten microliters of human milk were mixed with myristic-d27 as an internal standard. Protein precipitation was achieved by adding cold 80% MeOH (−20 °C). The sample was shaken for 10 min at 4 °C and then centrifuged at 20,000× *g* for 5 min at 4 °C. The supernatant was transferred to a tube and stored at 4 °C. A second extraction was performed on the pellet with 50 µL of water. After vortexing and centrifuging at 20,000× *g* for 5 min at 4 °C, the supernatants were combined, vortexed, and centrifuged again at 20,000× *g* for 5 min at 4 °C. The final supernatant was dried under a stream of nitrogen, and the samples were prepared for targeted metabolomics, as previously described [32,33]. Briefly, a two-step derivatization protocol was used, with methoxyamination, according to the method of Fiehn [34], followed by silylation with MTBSTFA/TBDMCS, according to the method of Patel [35]. The samples were analyzed on an Agilent 8890 GC equipped with a DB5-MS+DG capillary column coupled to an Agilent 5977B MS instrument operating under electron impact (EI) ionization at 70 eV (Agilent Technologies, Santa Clara, CA, USA) [36]. A 1 µL aliquot of each sample was injected in split mode at 250 °C, with helium as the carrier gas, and the flow rate of helium was 1 mL/min. The temperature of the GC was maintained at 50 °C for 2 min, before being raised to 150 °C (rate of 20 °C/min) for 5 min, and then to 300 °C (rate of 10 °C/min), and maintained at 300 °C for 10 min. The MS source and quadrupole were maintained at 230 °C and 150 °C, respectively, and the detector was operated in scanning mode for the mass range 50 to 650 Da at a signal rate of 5.1 scans/sec. Data analyses were performed using the Agilent MassHunter Workstation Software version 11.1 (Agilent Technologies, Santa Clara, CA, USA). Metabolites were identified post-deconvolution using the NIST/EPA/NIH Mass Spectral Library [37].

A total of 60 metabolites were initially analyzed in the human milk samples using GC/MS: acetylaspartate, acetylaspartic acid, adenine, alanine, aminoadipic acid, aminoisobutyric acid, aconitic acid, alpha-tocopherol, arachidonic acid, ascorbic acid, asparagine, aspartate, aspartic, citramalate, citrate, cholesterol, creatinine, cysteine, cytosine, edetic acid, fumarate, gamma-aminobutyric acid, glutamate, glutamine, glycine, glycerol, guanine, histidine, homoserine, homosalate, hydroxyproline, hypoxanthine, iminodiacetic acid, isoleucine, ketamine, lactate, leucine, lysine, malate, methionine, myristic acid, niacinamide, ornithine, oxaloacetate, phenylalanine, phosphoric acid, proline, pyroglutamate, serine, succinate, taurine, threonine, thymine, tryptophan, tyrosine, uracil, urea, uric acid, uridine, and valine. However, due to the limitations of the method, only 24 metabolites were accurately detected, while the remaining 36 metabolites were either undetected or had signals below the detection threshold. The 24 metabolites were selected for further analysis: amino acids (glutamate, proline, alanine, glycine, valine, leucine, isoleucine, methionine, serine, and phenylalanine), TCA cycle intermediates (succinate, malate, α-ketoglutarate, and citrate), fatty acids (myristic acid and arachidonic acid), and other metabolites related to energy metabolism (lactate, urea, glycerol, phosphoric acid, pyroglutamate, uracil, homosalate, and cholesterol).

### 2.5. Statistical Analyses

All statistical analyses were performed using R software version 2024.04.2+764. To compare the human milk metabolite profile between GDM+ and GDM- women, Student’s *t*-test for parametric variables and the Mann–Whitney U test for non-parametric variables were used. A Bonferroni correction was applied to adjust for multiple comparisons. Principal component analysis (PCA) and multiple factor analysis (MFA) were conducted using the FactoMineR and factoextra R packages. To explore global metabolite profiles while accounting for intercorrelations, we conducted a MANOVA on the PCA scores. Moreover, the MFA models included groups of variables, as follow: maternal adiposity [fat mass (kg), visceral adipose tissue volume (cm³); 2 variables], maternal glycemic profile [1-h post-OGTT insulinemia (pmol/L), 2-h post-OGTT glycemia (mmol/L), HOMA-IR; three variables], maternal lipid profile (triglycerides, chol/hdlc ratio; two variables), and milk composition [proteins (g/100 mL), triglycerides (g/100 mL), lactose (g/100 mL); three variables]. The variables included within the maternal lipid profile were selected based on their clinical relevance and their ability to capture significant lipid imbalances in a metabolic context. Plasma triglyceride levels are a key marker of metabolic disturbances associated with GDM, while the chol/hdlc ratio is a well-recognized indicator of cardiovascular risk and lipid abnormalities [38]. Spearman correlations were performed to study the association between maternal metabolic profile, human milk metabolites, and infant growth. Significance was determined using nominal p-values, with a threshold of *p* < 0.05. The models were adjusted for maternal age, and similar results were obtained. The results were thus presented without adjustments for age.

## 3. Results

Table 1 summarizes the characteristics of the mothers and infants. As previously reported by Dugas et al. in this cohort, maternal age, fat mass percentage, triglycerides, and chol/hdlc ratio were higher in the GDM+ group (*n* = 24) compared to the GDM- group (*n* = 29) [20]. At the 2-month postpartum visit, none of the participants in either the GDM+ or GDM- group had diabetes, and no differences in the glycemic profile were noted between the two groups, except for glucose AUC. Additionally, plasma HDLc levels and gestational age at birth were lower in the GDM+ group, and the infant sex distribution varied by GDM status. The weight-for-age z-score (WAZ) at 2 months was higher among GDM+ infants compared to GDM- infants.

To study the impact of GDM on the human milk metabolite profile, we performed targeted metabolomics of molecules involved in energy metabolism in 20 milk samples from GDM+ women and 29 milk samples from GDM- women. Unsupervised analyses were performed (Figure 1). First, PCA indicated that the metabolite profile of GDM+ human milk differed from that of GDM- human milk. The two groups were mostly distinct according to the analyzed metabolite profile (Figure 1A,B). Samples were separated mainly by the principal component 2. The major contributors to variability between human milk samples from GDM+ and GDM- were methionine, valine, glycerol, and myristic acid levels (Figure 1C,D). Additionally, complementary analyses were conducted to explore the impact of exclusive breastfeeding on milk composition, as well as the effect of participants’ fat mass percentages on milk biology. Supplementary figures show that excluding the two participants who did not exclusively breastfeed, or adjusting for the fat mass, led to similar results and conclusions (Appendix A).

Then, each metabolite was compared between the GDM+ and GDM- groups. In total, out of the 24 metabolites detected and quantified, 6 were found to be significantly different between GDM+ and GDM- human milk (Figure 2). Specifically, there were higher levels of glycerol, uracil, myristic acid, arachidonic acid, and cholesterol in GDM+ human milk. On the other hand, lower levels of pyroglutamate were observed in GDM+ human milk. However, this result was no longer significant after adjustment for multiple comparisons. These results suggest that GDM may also have long-term implications for the mammary gland and human milk composition.

The analysis of the correlation of metabolites among themselves highlighted two major groups of positively correlated metabolites (Figure 3). One included cholesterol, phosphoric acid, myristic acid, glycerol, and arachidonic acid, thus being comprised mostly of lipids. Uracil, linked to purinergic regulation pathways [39], was also part of this group. Importantly, five of the six metabolites identified to be significantly different between milk from GDM- and GDM+ mothers (Figure 2) were part of this group (Figure 3). The other group included mostly amino acids and energy metabolism intermediates, i.e., methionine, valine, serine, lactate, and malate. Finally, two metabolites, namely citrate and succinate, key intermediates of the TCA cycle [17], were not included in the former two groups and negatively correlated with the other metabolites. Altogether, these results indicate that two distinct pools of metabolites emerged according to the GDM status, suggesting that metabolites within each pool are possibly secreted together and associated with GDM.

To understand how maternal metabolic profiles, in addition to GDM status, were associated with human milk composition, a MFA was performed (Figure 4). It was hypothesized that maternal metabolic profiles, such as adiposity and glucose metabolism, could be correlated to the milk composition and possibly explain, at least in part, the differences observed between milk from GDM+ and GDM- women. We observed that the glycemic profile at 2 months postpartum was closely related to the human milk metabolite profile along dimension 2, suggesting a significant association between both variables (Figure 4A). Moreover, the first dimension of the MFA model showed that GDM status, milk composition, and human milk metabolite levels were correlated. The metabolites that most influenced the correlations with other variables along dimension 1 (Figure 4C) were myristic acid and uracil. Those which most influenced the correlations with maternal glycemic profile (Figure 4D) were methionine, α-ketoglutarate, proline, glutamate, valine, and serine. These results suggest a close link between GDM status, the maternal metabolic profile at 2 months postpartum, and the human milk composition.

Additionally, to better understand the relationship between each maternal factor and human milk composition, correlation analyses were performed between the maternal profile and each human milk metabolite (Figure 5A). Markers of insulin sensitivity, HOMA-IR fasting insulin, were positively correlated with succinate and malate. The mother’s glycemic profile was positively correlated with several metabolites, such as glutamate, α-ketoglutarate, valine, methionine, lactate, leucine, isoleucine, myristic acid, uracil, arachidonic acid, and glycerol. Negative correlations were found between the mothers’ lipid profiles and proline.

Correlation analyses were also performed separately in GDM+ women (Figure 5B) and GDM- women (Figure 5C) to assess whether associations differed according to the GDM status. For the GDM- group (Figure 5B), the lipid profile was positively correlated with citrate. Among GDM+ women, lipid profile was positively correlated with uracil, and cholesterol was negatively correlated with glycine and proline. Moreover, maternal blood cholesterol level was positively correlated with the level of cholesterol in human milk of GDM+ women (Figure 5C). The glycemic profile of GDM+ mothers was also positively correlated with succinate, malate, myristic acid, uracil, and arachidonic acid. Succinate was also, for the whole group (Figure 5A), positively correlated with the maternal metabolic profile. Separate correlation analyses for milk from GDM+ and GDM- mothers revealed specific associations between maternal glycemic and lipid profiles and human milk metabolites, supporting the hypothesis that the maternal metabolic status differentially influences milk composition depending on the GDM status.

Finally, we aimed to investigate whether human milk metabolites were correlated with infant growth. Positive correlations were observed in the entire cohort with WAZ at 2 months, including correlations with myristic acid, uracil, arachidonic acid, glycerol, and phosphoric acid (Figure 6A). Proline and urea levels were negatively correlated with WAZ at 2 months (Figure 6A). Correlation analyses were then conducted separately by GDM status. In human milk from GDM- mothers, malate, arachidonic acid, myristic acid, and phosphoric acid were positively correlated with infant growth (Figure 6B). However, proline and valine were negatively correlated with WLZ and WAZ at 2 months (Figure 6B). In human milk from GDM+ mothers, glycine and α-ketoglutarate levels were negatively correlated with WLZ and LAZ at 2 months, respectively (Figure 6C). In conclusion, several human milk metabolites were associated with infant growth, showing distinct patterns between milk from GDM- and GDM+ mothers.

## 4. Discussion

This study revealed significant differences between the human milk of GDM+ and GDM- women, with several metabolites, including glycerol, cholesterol, uracil, myristic acid and arachidonic acid, being present at higher levels in GDM+ milk, and pyroglutamate at lower levels. Furthermore, these metabolite differences were linked to maternal metabolic profiles, such as adiposity and glycemic profiles. Finally, human milk metabolite levels were correlated with infant growth outcomes, showing distinct patterns according to GDM status.

Maternal metabolic profile and infant growth were positively associated with many metabolites present in human milk, particularly lipid-related metabolites (arachidonic acid, myristic acid, cholesterol, and glycerol) and TCA cycle intermediates (malate, citrate, succinate, and α-ketoglutarate). We notably detected higher levels of arachidonic acid in GDM+ human milk, which is consistent with a study that showed high arachidonic acid levels in the plasma of pregnant women with GDM, which correlated also with insulin resistance [40]. Furthermore, this increase in arachidonic acid in GDM+ human milk is consistent with the results from a recent study on the endocannabinoidome system, where higher levels of the arachidonic acid derivative 2-MAG were observed in this cohort [21]. Additionally, myristic acid levels were found to be higher in human milk from GDM+ women compared to GDM- women. This fatty acid was correlated with infant growth in the GDM- group, but this correlation was not found in the GDM+ group. Myristic acid is a saturated fatty acid known to improve glycemia by increasing glucose absorption and reducing insulin resistance in peripheral tissues [41]. While a previous study reported lower levels of myristic acid in the human milk of obese women with GDM, our study observed higher levels of this fatty acid in the human milk of women with GDM [24]. This discrepancy may be explained by differences in study populations, such as variations in obesity status. We also found higher levels of glycerol and cholesterol in GDM+ human milk compared to that of GDM- women. GDM-related insulin resistance can lead to increased lipolysis, releasing glycerol, which may explain the higher glycerol levels observed in GDM+ milk [42,43]. The higher cholesterol level in GDM+ human milk may be due to altered cholesterol absorption and HDL maturation in GDM+ women [44]. The altered lipid profile of GDM+ mothers could also explain the elevated cholesterol levels in their milk. Studies highlight the importance of cholesterol uptake from human milk, compared to milk formula, in regulating endogenous cholesterol in adulthood [45,46]. However, the long-term effects of high cholesterol levels in human milk on infant growth remain unclear and require further research. The only metabolite to be significantly lower in GDM+ milk was pyroglutamate. Pyroglutamate is a derivative of the amino acid glutamate (glutamic acid) and has been observed to be decreased in people with type 2 diabetes [47]. In the literature, decreased levels of pyroglutamate have been observed in the serum of pregnant women with GDM+ compared to GDM- control women [23]. Since pyroglutamate is a component of the glutathione cycle, its decreased level in GDM+ milk could reflect a disturbed maternal metabolic state (oxidative stress).

Moreover, another group of metabolites positively impacted by the maternal metabolic profile comprised TCA cycle intermediates. Indeed, succinate was not significantly different between the GDM+ and GDM- groups. However, it was positively correlated with anthropometric maternal data and markers of glycemic control (such as HOMA-IR, fasting insulin, insulin at 60 min and 120 min, glucose at 60 min, and HbA1c) when combining all the women or when specifically studying the GDM- group. This suggests that maternal metabolic profiles other than GDM can influence succinate levels in human milk. In the GDM+ group, higher succinate levels were associated with poorer glycemic control. Succinate plays a key role in the TCA cycle, contributing to cellular energy production, and it has been associated with reduced white adipose tissue in obesity and insulin resistance and increased respiratory capacity due to higher mitochondrial content [48]. Moreover, succinate is a ligand for the SUCNR1 receptor, whose activation leads to insulin secretion [49]. Chronically elevated levels of succinate have been associated with metabolic disorders such as diabetes [49].

The correlation profiles between human milk metabolites and infant growth differed between the GDM- and GDM+ groups. In GDM+ human milk, glycine and α-ketoglutarate were negatively correlated with WLZ and LAZ at 2 months, respectively. These correlations were not found in the GDM- group. α-ketoglutarate is a key intermediary in the TCA cycle, connecting energy metabolism to biosynthetic pathways, and is linked to glutamate and glutamine metabolism, which are essential for growth and development [23,50]. A study showed that low concentrations of α-ketoglutarate stimulate cell growth by improving glucose and glutamine metabolism [51]. If the infant does not receive enough α-ketoglutarate through human milk, this could lead to an imbalance in the TCA cycle, potentially affecting growth. Moreover, α-ketoglutarate has been shown to inhibit the mTOR pathway, which is central to cell growth and autophagy [52]. Thus, lower levels of α-ketoglutarate in human milk may disrupt mTOR balance, potentially leading to hyperactivation of mTOR and promoting excessive cell growth while inhibiting autophagy. More studies are required to fully understand the impact of α-ketoglutarate on mammary gland biology. Similarly to α-ketoglutarate, glycine levels in human milk were negatively correlated with infant growth at 2 months in the GDM+ group. Glycine is an essential precursor for several metabolic pathways, including glutathione synthesis and energy metabolism [53]. It has also been consistently found at lower levels in individuals with metabolic disorders such as obesity and type 2 diabetes [12,54,55,56]. In our study, there was a trend toward reduced glycine concentration in GDM+ human milk (Figure 2), which could indicate that maternal metabolic imbalances may influence milk composition. This metabolic imbalance could impact infant growth processes through pathways that remain poorly understood. Indeed, the role of glycine in growth has yet to be explored, and its metabolic imbalance could cause disturbances in cell growth and energy production [57]. Thus, our results suggest that the composition of human milk in glycine and α-ketoglutarate, influenced by maternal metabolism, could have repercussions on the infant’s energy metabolism. During early infancy, human milk provides essential nutrients, including amino acids and organic acids, that support rapid growth and brain development. Alterations in these nutrient profiles, as observed in GDM+ human milk, may result in metabolic imbalances that influence infant growth trajectories during this critical window of development.

The type of treatment used to manage GDM was not built into the current study design, which is a limitation. However, some additional analyses were performed where human milk metabolite profiles from GDM-, GDM+ (treated with insulin or hypoglycemic agent), and GDM+ (treated with diet only) were compared (Appendix A). The metabolite profile of milk from GDM- women separated better from GDM+ women treated with hypoglycemic agents than those treated with diet alone. However, the profiles of the two GDM+ groups did not differ significantly by treatment. Without treatment adjustment, LDA showed a 4% error rate, correctly classifying 96% of patients based on metabolite profiles, adjusting for treatment-improved performance and reducing the error rate to 0% (Appendix A). However, this variable is not essential for classification, as the model remains highly effective even without it. Given the small sample size within each group, further studies are needed to investigate the role of treatment and/or severity of GDM on human milk composition.

The current study has other limitations, such as analyzing milk samples only at 2 months postpartum, as well as a relatively small sample size of participants. Tracking changes in the metabolite profile in the milk and monitoring infant growth over the subsequent months would have provided additional valuable insights. Additionally, the rate of exclusive breastfeeding differed between the GDM+ and GDM- groups, which could also influence the human milk metabolite profile. An analysis was carried out excluding the two participants in the GDM+ group who had not exclusively breastfed, and similar results were found. Although breastfeeding practices differed among our participants, the results remained similar. Also, the findings cannot be generalized to other populations, as the cohort primarily consisted of well-educated Caucasian women. Therefore, further studies with larger sample sizes are necessary, along with more diverse populations of women. Despite these limitations, this study had several strengths. This is the first study, to our knowledge, to explore the relationship between the metabolite profile of human milk and GDM, as well as its connection to infant growth among those exposed to GDM in utero. Additionally, the study strengths include the simultaneous collection of human milk samples alongside infant growth measurements and assessments of maternal adiposity and cardiometabolic health during the 2-month postpartum visit.

## 5. Conclusions

In summary, this study investigated the impact of GDM and the maternal metabolic profile on the human milk metabolite profile and how specific human milk metabolites were associated with infant growth. Results revealed significant differences between the human milk metabolite profile of GDM+ and GDM- women, with higher levels of several metabolites, including glycerol, cholesterol, uracil, myristic acid, and arachidonic acid, and lower levels pyroglutamate, in GDM+ women. We also observed that maternal adiposity and glycemic profiles were associated with these human milk metabolites, with some metabolites also being correlated with infant growth. These correlations show that, despite the GDM status, other maternal postpartum factors, such as lipid profile or blood glucose level, influence infant growth through the human milk metabolite profile. Altogether, while validation is required, these results highlight that GDM is associated with changes in the metabolite composition of human milk, which is, in turn, linked to infant growth at 2 months.

## Figures and Tables

**Figure 1 nutrients-17-01466-f001:**
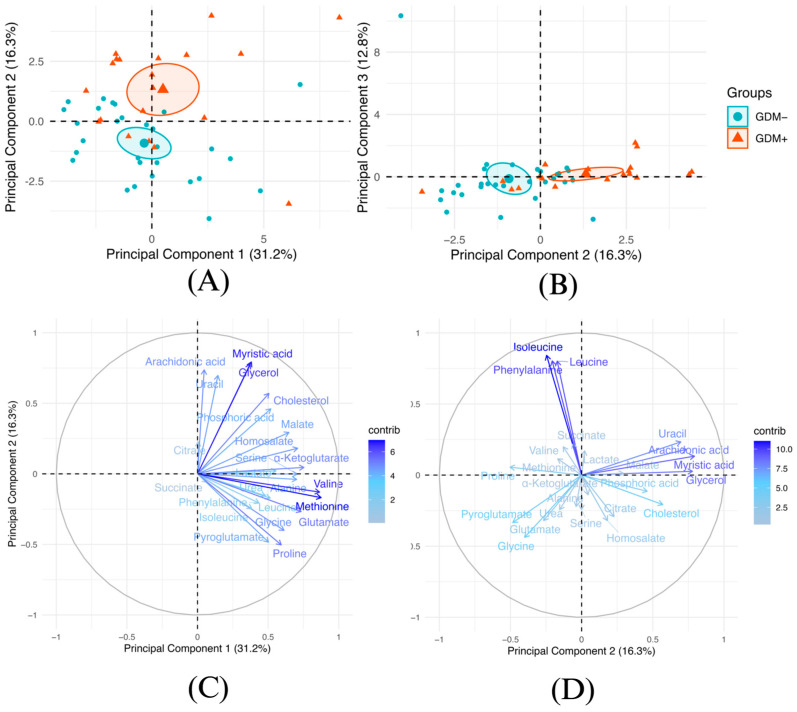
Distinct metabolite profiles in human milk from GDM+ and GDM- mothers. Visualization of the principal component analysis (PCA) modelling the human milk metabolite profile. The loading plot of all human milk metabolites in principal components 1 and 2 (**A**) or principal components 2 and 3 (**B**). Ellipses are confidence intervals from the mean center of each group (GDM-: Blue, GDM+: Red). Graph of individuals of principal components 1 and 2 (**C**) and of principal components 2 and 3 (**D**) of the PCA model.

**Figure 2 nutrients-17-01466-f002:**
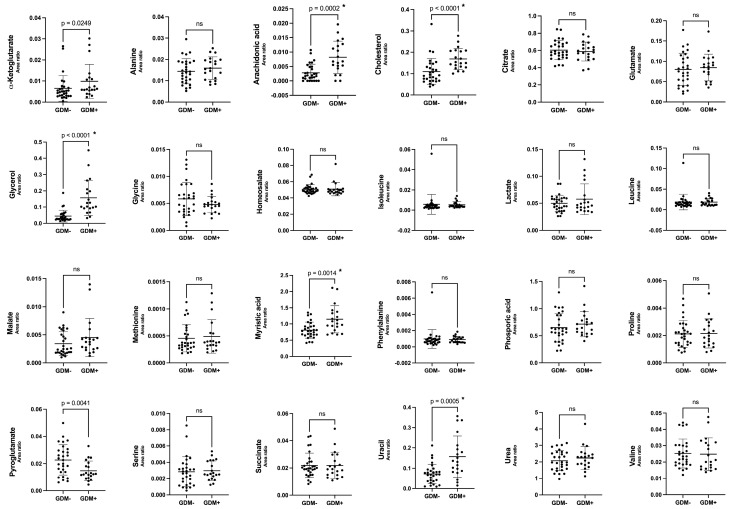
Specific metabolite levels are higher in human milk of GDM+ women. Boxplots include the mean and standard deviation. When significant, *p*-value is indicated before Bonferroni correction. * indicates significant *p*-value after adjustment for Bonferroni; n.s.: not significant.

**Figure 3 nutrients-17-01466-f003:**
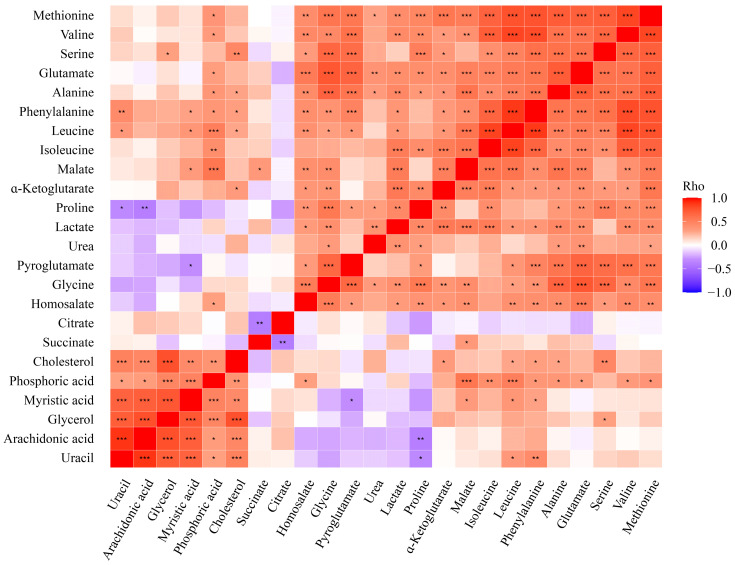
Profile of correlations between human milk metabolites. Heatmap of Spearman correlation between the 24 metabolites studied in all human milk samples from GDM+ and GDM- women. Metabolites are arranged by cluster (based on their correlation similarity). * *p* value < 0.05, ** *p* value < 0.005, and *** *p* value < 0.001.

**Figure 4 nutrients-17-01466-f004:**
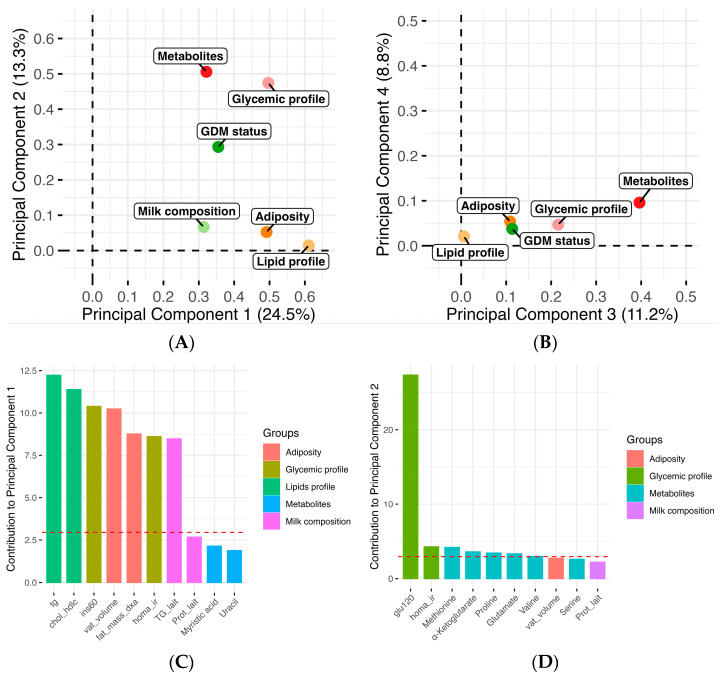
Maternal metabolic profile and human milk composition are associated with GDM status. Visualization of the multiple factor analysis (MFA) modelling human milk metabolite with variables of adiposity (fat mass, visceral adipose tissue), glycemic (insulin, glycemia, HOMA-IR) and lipid (triglycerides, chol/HDLc ratio) profiles, milk composition (protein, lactose, triglycerides), and GDM status. Graphs showing the factors contributing to principal components 1 and 2 (**A**), and to principal components 3 and 4 (**B**), of the MFA model are presented. Graphs highlighting the variables that contribute the most to principal component 1 (**C**) and to principal component 2 (**D**) are also displayed. The red line represents the expected average contribution of each variable to the component, assuming equal contribution. Variables above the red line contribute more than average to the component.

**Figure 5 nutrients-17-01466-f005:**
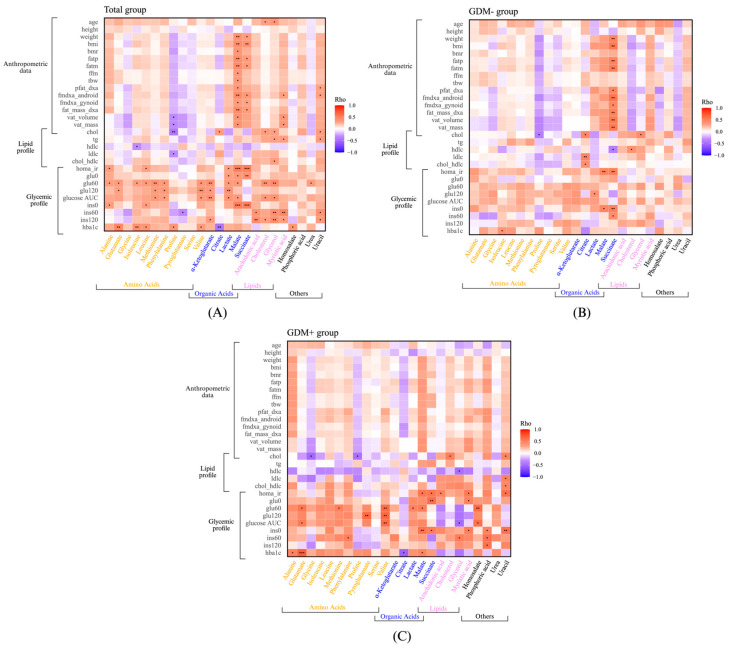
The maternal metabolic profile is associated with human milk metabolites differently in the GDM- and GDM+ groups. Heatmap of Spearman correlation between human milk metabolites and maternal data in total group (**A**), in GDM- group (**B**), and in GDM+ group (**C**). * *p* value < 0.05, ** *p* value < 0.005, and *** *p* value < 0.001. BMI: body mass index; BMR: basal metabolic rate; FATP: fat percentage; FATM: fat mass; FFM: fat free mass; TBW: total body water; Pfat_dxa: fat mass percentage; Fmdxa_android: percentage of fat mass in the android region; Fmdxa_gynoid: percentage of fat mass in the gynoid region; Fat_mass_dxa: fat mass with dexa method; Vat_volume: visceral adipose tissue volume; Vat_mass: visceral adipose tissue mass; Hdlc: high-density lipoprotein cholesterol; Hba1c: glycated hemoglobin; Glucose AUC: glucose area under the curve; Ldlc: low-density lipoproteins cholesterol; Chol/hdlc: cholesterol to Hdlc ratio.

**Figure 6 nutrients-17-01466-f006:**
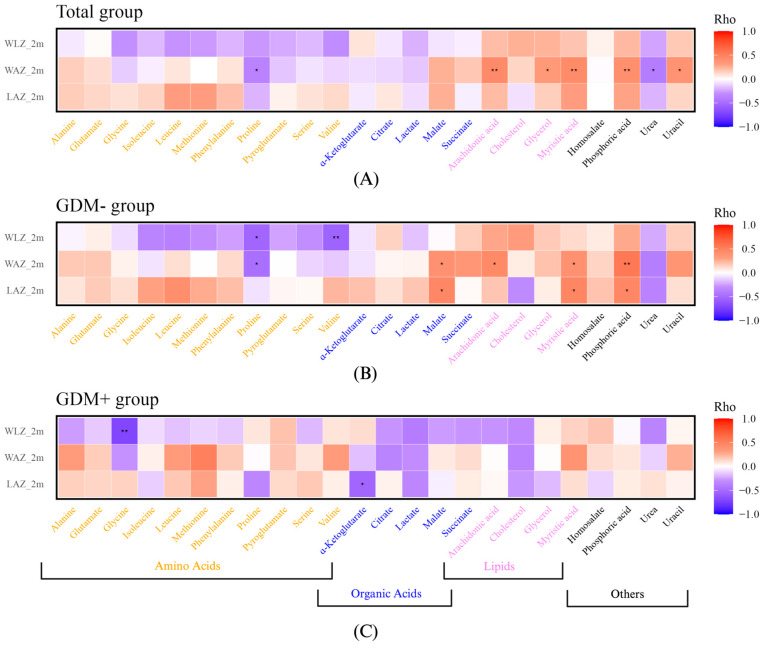
Human milk metabolites are associated with infant growth differently in the GDM- and GDM+ groups. Heatmaps of Spearman correlations between human milk metabolites and infant growth in total group (**A**), GDM- group (**B**), and GDM+ group (**C**). * *p* value < 0.05 and ** *p* value < 0.005. LAZ: length-for-age z-score; WAZ: weight-for-age z-score; WLZ: weight-for-length z-score; 2m: at 2 months old.

**Table 1 nutrients-17-01466-t001:** Characteristics of mothers and infants in the GDM- and GDM+ group.

	GDM- (*n* = 29)	GDM+ (*n* = 24)	*p* Value
** Maternal**			
Metabolic data			
Age (years)	30.0 ± 3.1	33.6 ± 3.6	<0.001 ***
Height (m)	1.64 ± 0.07	1.65 ± 0.04	0.60 ^A^
Weight (Kg)	74.6 ± 17.6	84.8 ± 19.7	0.067
BMI (kg/m^2^)	27.7 ± 5.9	31.2 ± 7.1	0.07
Fat mass (%)	38.0 ± 8.6	42.8 ± 6.8	0.04 *
Fasting glucose (mmol/L)	4.80 ± 0.31	5.04 ± 0.42	0.05
Glucose AUC (mmol.min/L)	670 ± 101	804 ± 118	<0.001 ***
Fasting insulin (pmol/L)	51.0 ± 35.34	50.9 ± 20.5	0.64 ^A^
Hba1c (%)	5.18 ± 0.37	5.31 ± 0.24	0.14 ^A^
Cholesterol (mmol/L)	5.09 ± 1.04	5.24 ± 1.13	0.62
Triglycerides (mmol/L)	0.84 ± 0.40	1.15 ± 0.46	0.007 **^A^
Hdlc (mmol/L)	1.74 ± 0.37	1.56 ± 0.29	0.04 *
Ldlc (mmol/L)	2.96 ± 0.90	3.15 ± 1.05	0.423
Chol/hdlc	3.02 ± 0.80	3.45 ± 0.94	0.047 *
GDM treatment			
Diet only	—	9 (45%)	—
Insulin or oral hypoglycemic agent	—	11 (55%)	—
Gestational age (weeks)	39.41 ± 1.1	38.6 ± 1.0	0.01 *
Breastfeeding			0.23 ^B^
Exclusive	29 (100%)	22 (91.7%)	
Non-exclusive	0 (0%)	2 (8.3%)	
Timing of human milk collection			0.28
Day (6:00 a.m. to 6:00 p.m.)	25 (86%)	14 (70%)	
Night (6:00 p.m. to 6:00 a.m.)	4 (14%)	6 (30%)	
Delivery			1 ^B^
Vaginal Birth	27 (93.1%)	22 (91.7%)	
Cesarean	2 (6.9%)	2 (8.3%)	
** Infant**			
Sex			0.01 *^C^
Boys	12 (41%)	17 (74%)	
Girls	17 (59%)	6 (26%)	
Length (cm)			
Birth	50.70 ± 1.60	50 ± 4.17	0.42
2 months	57.63 ± 1.73	57.77 ± 2.95	0.86
Weight (kg)			
Birth	3.34 ± 0.38	3.37 ± 0.34	0.75
2 months	5.06 ± 0.71	5.48 ± 0.78	0.08
LAZ			
Birth	0.68 ± 0.88	0.64 ± 0.91	0.89
2 months	0.04 ± 0.83	0.17 ± 1.07	0.66
WAZ			
Birth	0.10 ± 0.84	0.11 ± 0.69	0.98
2 months	−0.32 ± 0.88	0.23 ± 0.72	0.03 *
WLZ			
Birth	−0.64 ± 1.26	−0.52 ±1.42	0.79
2 months	−0.33 ± 1.07	0.04 ± 0.73	0.24

The data are presented as means ± the standard deviation. * *p* value < 0.05, ** *p* value < 0.005, and *** *p* value < 0.001 by Student *t* test, Wilcoxon test ^A^, Fisher’s exact test ^B^, or Chi-square test ^C^, between group of GDM status. BMI: body mass index; Hdlc: high-density lipoproteins cholesterol; Hba1c: glycated hemoglobin; Glucose AUC: glucose area under the curve; Ldlc: low-density lipoprotein cholesterol; Chol/hdlc: cholesterol to Hdlc ratio; LAZ: length-for-age z-score; WAZ: weight-for-age z-score; WLZ: weight-for-length z-score. This table is based on a previously described cohort [20].

## Data Availability

The datasets used and/or analyzed during the current study are available from the corresponding author upon reasonable request.

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
