# Peer review of "Variations in Human Milk Metabolites After Gestational Diabetes: Associations with Infant Growth"

_nutrients, 2025, doi:10.3390/nu17091466_

Round 1

Reviewer 1 Report

Comments and Suggestions for Authors

This was an interesting read. The authors provide a strong background section. However, I disagree with the following statement: "However, the extent to which 93 GDM affects other milk nutrients is still largely unknown. Currently, there is insufficient data on how GDM influences human milk composition and its impact on infant health." There are over 50 studies looking at GDM and milk composition. I suggest making this sentence more specific to your research question gaps.

I thought the objectives could be made clearer. The title and objectives at the end of the background section highlight the focus on human milk composition based on glycemic state, and how these human milk components subsequently relate to infant growth. It seems there were many more exploratory questions included in the manuscript, including linking lipids with the human milk components. Please ensure the objectives are all clearly listed with what is presented in results.

The methodology states that milk was thawed then aliquoted. This counters standard practice of not thawing milk until ready for analysis. Please cite literature for the specific compounds explored and effects of thawing. 

In addition, I have concerns about the statistical analysis. The relatively small groups may be too small for high-dimensional analyses like PCA or MFA, which can lead to overfitting or unstable results. In addition, the use of Student’s t-test or Mann–Whitney U test for each metabolite did not mention multiple comparison correction (e.g., Bonferroni or FDR). This increases the risk of false positives (Type I errors) and needs to be done. Finally, these tests assume variables are independent, but metabolites are often correlated (as the authors found) due to shared pathways. Univariate tests might not capture these interdependencies well. MANOVA or Multivariate Generalized Linear Models like MaAsLin2 may be better options.

Importantly, the adjusted models were explored with maternal age but not with BMI, diet, physical activity, time of milk collection, which are all important confounders for milk composition and metabolome. Lack of diet has an added concern in that GDM+ and GDM- were recruited at different time points (2017-2019 vs 2020), which may have led to temporal dietary trends, change in dietary practices, etc. The authors also found maternal adiposity to be correlated with human milk components but this again was not adjusted for which makes it challenging to conclude the link between GDM directly and respective components.

Finally, the selection of variables for multiple factor analysis (MFA) (e.g., specific maternal adiposity, glycemic, and lipid profile markers) was based on clinical relevance, but the exclusion of other potentially relevant variables might limit the comprehensiveness of the analysis.

Thank you, and look forward to a revised version.

Author Response

Response to reviewers

First, we would like to sincerely thank the reviewers for their careful analysis of our manuscript. Their comments and suggestions have been carefully considered in the revised version, as explained below. We are convinced that these adjustments have strengthened the overall quality of the manuscript. All changes have been clearly highlighted in the document.

Reviewer 1

  1. This was an interesting read. The authors provide a strong background section. However, I disagree with the following statement: "However, the extent to which GDM affects other milk nutrients is still largely unknown. Currently, there is insufficient data on how GDM influences human milk composition and its impact on infant health." There are over 50 studies looking at GDM and milk composition. I suggest making this sentence more specific to your research question gaps.

The sentence was changed accordingly. The new sentence, lines 239-241, reads as follow : « However, few have focused on the metabolic profile of milk, particularly metabolites and bioactive molecules related to energy metabolism, and how these may influence infant growth trajectories. Currently, there is insufficient data on how GDM influences human milk composition and its impact on infant health. »

  1. I thought the objectives could be made clearer. The title and objectives at the end of the background section highlight the focus on human milk composition based on glycemic state, and how these human milk components subsequently relate to infant growth. It seems there were many more exploratory questions included in the manuscript, including linking lipids with the human milk components. Please ensure the objectives are all clearly listed with what is presented in results.

A sentence was added to broaden the objectives and incorporated all maternal data that we analyze for correlations with metabolites.

Line 246-248 : « We also aimed to explore how the maternal metabolic profile (anthropometric, glycemic and lipid profiles) is associated with human milk components. »

  1. The methodology states that milk was thawed then aliquoted. This counters standard practice of not thawing milk until ready for analysis. Please cite literature for the specific compounds explored and effects of thawing. 

A meta-analysis shows that freezing and thawing human milk does not result in significant changes in its macronutrient and energy composition [1]. This review has identified that the data on the effects of milk thawing and warming is limited and often contradictory.Nevertheless, our protocol aligns with recommended practices to minimize alterations to sensitive components of human milk, and all samples were treated in the same way [2]. Also, all participants received written instructions for sample collections.

  1. In addition, I have concerns about the statistical analysis. The relatively small groups may be too small for high-dimensional analyses like PCA or MFA, which can lead to overfitting or unstable results.

We acknowledge the reviewer's concern regarding sample size in high-dimensional analyses. However, we would like to point out that principal component analysis (PCA) has been effectively applied in metabolomics studies with comparable or even smaller sample sizes, demonstrating the method's applicability in small-sample contexts[3]. In addition, a study demonstrated the validity of the use of PCA on small sample sizes [4].

Moreover, the MFA was performed using 6 groups of clinically related variables (metabolites, adiposity, glycemic, lipid profile, milk composition, and GDM status), each group containing 1 to 3 variables. While our sample is relatively small, the number of variables included in the MFA was limited (12 variables), allowing for a reasonable ratio of variable-to-individual. To support the results obtained through MFA, correlation analyses were performed between the maternal profile and each human milk metabolite.

In addition, the use of Student’s t-test or Mann–Whitney U test for each metabolite did not mention multiple comparison correction (e.g., Bonferroni or FDR). This increases the risk of false positives (Type I errors) and needs to be done.

We thank you for this comment. Bonferroni adjusment was made and similar results were obtained.  Figure 2 in the revised manuscrit now includes result before and after adjustment for multiple comparisons.

Finally, these tests assume variables are independent, but metabolites are often correlated (as the authors found) due to shared pathways. Univariate tests might not capture these interdependencies well. MANOVA or Multivariate Generalized Linear Models like MaAsLin2 may be better options.

However, while our sample size limits the use multivariate models such as MaAsLin2 or full MANOVA on metabolite levels, we did perform a MANOVA on the PCA scores. By testing group differences on these principal components, we partially address the interdependencies. We have now clarified this point in the Methods.

Lines 493-494 : « To explore global metabolite profiles while accounting for intercorrelations, we conducted a MANOVA on the PCA scores »

  1. Importantly, the adjusted models were explored with maternal age but not with BMI, diet, physical activity, time of milk collection, which are all important confounders for milk composition and metabolome. Lack of diet has an added concern in that GDM+ and GDM- were recruited at different time points (2017-2019 vs 2020), which may have led to temporal dietary trends, change in dietary practices, etc. The authors also found maternal adiposity to be correlated with human milk components but this again was not adjusted for which makes it challenging to conclude the link between GDM directly and respective components.

We understand the importance of variables such as body mass index (BMI), diet, physical activity, and timing of milk collection, and we recognize that these variables can influence milk composition and the metabolome. However, due to our small sample size, it was difficult to adjust for all of these variables simultaneously.

We therefore chose to focus on the variables most strongly associated with our results, such as maternal age, which is well documented as a factor influencing milk composition. We also explored additional adjustments for variables such as maternal fat mass and exclusive breastfeeding, which were significantly different between groups. The results obtained after these adjustments remained similar, as shown in the supplementary figures.

Although the nutritional component was not included in this study, and we acknowledge this as a limitation, previous research has shown that maternal diet does not always directly translate into variations in human milk composition. For instance, Marc et al. observed that maternal supplementation with DHA (an omega-3 fatty acid) does not consistently lead to increased DHA levels in breast milk [5]. Furthermore, the concentration of certain nutrients, such as proteins and lactose, is generally stable in human milk and minimally influenced by the mother's diet [6,7].

While some components of human milk may vary based on dietary intake, others remain stable, making it challenging to study their association with milk composition in the context of GDM.

  1. Finally, the selection of variables for multiple factor analysis (MFA) (e.g., specific maternal adiposity, glycemic, and lipid profile markers) was based on clinical relevance, but the exclusion of other potentially relevant variables might limit the comprehensiveness of the analysis.

Given the relatively small sample size in our study, the most relevant variables were prioritized without compromising the robustness of the analysis.

  1. Yochpaz, S.; Mimouni, F.B.; Mandel, D.; Lubetzky, R.; Marom, R. Effect of Freezing and Thawing on Human Milk Macronutrients and Energy Composition: A Systematic Review and Meta-Analysis. Breastfeeding Medicine 2020, 15, 559–562, doi:10.1089/bfm.2020.0193.
  2. Stinson, L.F.; George, A.; Gridneva, Z.; Jin, X.; Lai, C.T.; Geddes, D.T. Effects of Different Thawing and Warming Processes on Human Milk Composition. The Journal of Nutrition 2024, 154, 314–324, doi:10.1016/j.tjnut.2023.11.027.
  3. Shaukat, S.S.; Rao, T.A.; Khan, M.A. Impact of Sample Size on Principal Component Analysis Ordination of an Environmental Data Set: Effects on Eigenstructure. Ekológia (Bratislava) 2016, 35, 173–190, doi:10.1515/eko-2016-0014.
  4. Yata, K.; Aoshima, M. Principal Component Analysis Based Clustering for High-Dimension, Low-Sample-Size Data 2015.
  5. Marc, I.; Piedboeuf, B.; Lacaze-Masmonteil, T.; Fraser, W.; Mâsse, B.; Mohamed, I.; Qureshi, M.; Afifi, J.; Lemyre, B.; Caouette, G.; et al. Effect of Maternal Docosahexaenoic Acid Supplementation on Bronchopulmonary Dysplasia–Free Survival in Breastfed Preterm Infants: A Randomized Clinical Trial. JAMA 2020, 324, 157, doi:10.1001/jama.2020.8896.
  6. Dias, B.; Nakhawa, D. Effect of Maternal Nutritional Status on the Biochemical Composition of Human Milk. Int J Res Med Sci 2016, 4541–4543, doi:10.18203/2320-6012.ijrms20163325.
  7. Lactose Intolerance: Myths and Facts. An Update. Arch Argent Pediat 2022, 120, doi:10.5546/aap.2022.eng.59.

Reviewer 2 Report

Comments and Suggestions for Authors

The authors present a cross-sectional study based on a small cohort of women with and without gestational diabetes mellitus (GDM), selected according to specific criteria (singleton and term pregnancies). GDM+ women were previously enrolled in an 18-month intervention study starting two months postpartum and were included in this cross-sectional study. The study investigates the impact of GDM and maternal metabolic status on the human milk metabolite profile, examining associations between specific milk metabolites and both maternal glycemic/lipid profiles and infant anthropometric data at 2 months postpartum.

The authors showed that gestational hyperglycemia may influence the composition of human milk beyond delivery, particularly increasing levels of specific lipid species (e.g. myristic acid, arachidonic acid, cholesterol), which could impact infant growth.

The manuscript presents several interesting findings, but a number of points require clarification or further elaboration:  

  1. Abstract (lines 33-34): The sentence “These metabolites correlated with maternal glycemic and lipid profiles and infant growth, differently according to the GDM status” (lines 33-34) is not very informative. Please clarify which metabolites showed differential correlations and how these differed by GDM status.
  2. Introduction (lines 58-59): It would be valuable to briefly discuss the reproductive hormonal changes (g. estrogen, progesterone, prolactin) that govern mammary gland development, particularly in the context of GDM. This would provide relevant physiological background without diverting from the main focus of the manuscript.
  3. Introduction: The discussion on the endocannabinoid system would benefit from more detail. Which specific metabolites were altered in GDM? A more in-depth discussion of the links between arachidonic acid and endocannabinoid-related pathways would strengthen the interpretation of the results.  
  4. Milk collection protocol (line 157): The milk sampling procedure should be more clearly described. Was the milk homogenized prior to freezing at -80°C to prevent phase separation upon thawing? Given the known diurnal variation in milk composition, especially for lipid content, how was this variability controlled (g. time of day)?
  5. OGTT data (line 134): Why was the area under the curve (AUC) for glucose not analyzed during the 75 g 2-hour oral glucose tolerance test? This could provide a more comprehensive assessment of maternal glycemic status?
  6. Metabolomics (lines 163-166): It is unclear whether a targeted or untargeted metabolomics approach was used. The use of external standards is not clearly described in the material and the expression of levels as area ratios (e.g. in Figure 2) suggests a semi-quantitative approach. Please clarify. Additionally, what do the ‘method limitations” mentioned in line 195 refer to? Please specify the detection thresholds and whether quality control –(QC) samples were used, and is so, which type of QC?
  7. Statistics (204-223): Please indicate which univariate tests were applied in Table 1. Also, in line 221, maternal BMI and infant birth weight, both potentially important confounding variables, were not adjusted for. Please justify the decision or consider adjusting for them.
  8. Results/Table 1: Please cite the original study referenced in Table 1 with the table legend. Furthermore, considering the sex-related differences in both groups (GDM- and GDM+), and the known influence of infant sex on milk composition, it would be valuable to address or at least explore this factor in the analysis or discussion.
  9. Figure 4 (line 324): What does the red line represent? If it indicates a significance threshold, how do metabolites such as myristic acid and uracil relate to this line?
  10. Figure 5: The variable labels are difficult to read. Please improve readability, and specify which variables refer to maternal and which to infant data.
  11. Figure S1: The meaning of the p-value in the PCA analysis is unclear, especially since PCA is an unsupervised method. Please clarify.
  12. Discussion- pyroglutamate: The discussion of pyroglutamate variability could be strengthened by addressing its role in glutathione metabolism. Is there literature connecting pyroglutamate or glutathione deficiency to GDM?
  13. Discussion-milk cholesterol: Several studies have suggested beneficial effects of cholesterol in breast milk compared to formula. A brief discussion of this relation to your findings would add valuable context.

Author Response

Response to reviewers

First, we would like to sincerely thank the reviewers for their careful analysis of our manuscript. Their comments and suggestions have been carefully considered in the revised version, as explained below. We are convinced that these adjustments have strengthened the overall quality of the manuscript. All changes have been clearly highlighted in the document.

Reviewers 2

The authors present a cross-sectional study based on a small cohort of women with and without gestational diabetes mellitus (GDM), selected according to specific criteria (singleton and term pregnancies). GDM+ women were previously enrolled in an 18-month intervention study starting two months postpartum and were included in this cross-sectional study. The study investigates the impact of GDM and maternal metabolic status on the human milk metabolite profile, examining associations between specific milk metabolites and both maternal glycemic/lipid profiles and infant anthropometric data at 2 months postpartum.

The authors showed that gestational hyperglycemia may influence the composition of human milk beyond delivery, particularly increasing levels of specific lipid species (e.g. myristic acid, arachidonic acid, cholesterol), which could impact infant growth.

The manuscript presents several interesting findings, but a number of points require clarification or further elaboration:  

  1. Abstract(lines 33-34): The sentence “These metabolites correlated with maternal glycemic and lipid profiles and infant growth, differently according to the GDM status” (lines 33-34) is not very informative. Please clarify which metabolites showed differential correlations and how these differed by GDM status.

The abstract has been revised in response to this comment.

Lines 31-36 : « Specific human milk metabolites showed distinct correlations with maternal glycemic as well as infant growth, depending on GDM status. While,  maternal glycemia was associated with succinate and malate in all groups, maternal glycemia was specifically correlated with valine and glutamate in GDM+ mothers. Besides, in GDM+ women, α-ketoglutarate and glycine were negatively correlated with infant growth.»

  1. Introduction (lines 58-59): It would be valuable to briefly discuss the reproductive hormonal changes (estrogen, progesterone, prolactin) that govern mammary gland development, particularly in the context of GDM. This would provide relevant physiological background without diverting from the main focus of the manuscript.

Thank you for this helpful suggestion. We recognize that reproductive hormonal changes, particularly those related to estrogen, progesterone, and prolactin, play a crucial role in mammary gland development, particularly during pregnancy. With regard to gestational diabetes mellitus (GDM), these hormones can be altered, thus influencing milk composition and breast function. In the revised version, wa have added a sentence to raise this point

Lines 81-87 : « Hormones play key roles in this process. They promote the development of mammary alveoli, increase transcription of milk component genes, stimulate the closure of epithelial cell tight junctions during early lactation, and mediate milk synthesis in response to infant suckling [1]. In the context of GDM, alterations in maternal metabolic and hormonal environments may influence the levels or actions of these hormones, potentially impacting mammary function and milk composition [2]. »

  1. Introduction: The discussion on the endocannabinoid system would benefit from more detail. Which specific metabolites were altered in GDM? A more in-depth discussion of the links between arachidonic acid and endocannabinoid-related pathways would strengthen the interpretation of the results.  

A sentence was added in the discussion to link this manuscript to the one on the endocannabinoidome system.

Lines 747-749 : « Furthermore, this increase in arachidonic acid in GDM+ human milk is consistent with the results from a recent study on the endocannabinoidome system where higher levels of the arachidonic acid derivative 2-MAG were observed in this cohort [3]. »

  1. Milk collection protocol(line 157): The milk sampling procedure should be more clearly described. Was the milk homogenized prior to freezing at -80°C to prevent phase separation upon thawing? Given the known diurnal variation in milk composition, especially for lipid content, how was this variability controlled ( time of day)?

Thank you for your comments. Changes have been made to the method section for human milk protocol. Homogenization was indeed carried out before aliquoting them. Furthermore, we acknowledge that human milk composition can vary during the day but this information was not taking into account in the study. However,  there was no difference according to the GDM status (Table 1).

Lines 341-349 : « Women were provided written instructions to collect 30 to 60 mL of human milk at the end of a feeding session. They recorded the date and time of collection and stored the milk in sterile cups in their home freezers. The frozen samples were then transported to the research center in insulated bags containing ice packs. All human milk samples were then stored at -80°C afterward. Frozen samples were defrosted at 0°C to 4°C, and whole human milk samples were vortexed at high speed for 30 seconds to ensure sample homogeneity immediately before aliquoting .and stored at -80 °C until batch analysis The composition of the milk, including its energy, triglyceride, lactose, and protein content, had been previously reported »

  1. OGTT data (line 134): Why was the area under the curve (AUC) for glucose not analyzed during the 75 g 2-hour oral glucose tolerance test? This could provide a more comprehensive assessment of maternal glycemic status?

We thank the reviewer for this valuable suggestion. We calculated the area under the curve (AUC) for glucose during the OGTT using the trapezoidal method. Indeed, AUC was statistically different between the two groups (p < 0.001) and this finding has been included in table 1 and in figure 5.

  1. Metabolomics (lines 163-166): It is unclear whether a targeted or untargeted metabolomics approach was used. The use of external standards is not clearly described in the material and the expression of levels as area ratios (e.g. in Figure 2) suggests a semi-quantitative approach. Please clarify. Additionally, what do the ‘method limitations” mentioned in line 195 refer to? Please specify the detection thresholds and whether quality control –(QC) samples were used, and is so, which type of QC?

Thank you for your comment. We confirm that an untargeted metabolomics approach was used. Absolute quantification was not performed, as no calibration curves were used. Internal standards were used for normalization between samples and the results provided relative quantification. We did not perform specific quantification curves using human milk as the biological matrice and, consequently, did not determine the lowest limit of detection (this is why no detection thresholds were defined). The mention of “method limitations” refers specifically to these aspects.

  1. Statistics(204-223): Please indicate which univariate tests were applied in Table 1. Also, in line 221, maternal BMI and infant birth weight, both potentially important confounding variables, were not adjusted for. Please justify the decision or consider adjusting for them.

Thank you for this comment. The univariate tests used in Table 1 are described in the legend. Also, we chose not to adjust for BMI and infant birth weight in the final models as they were not significantly different between groups.

However, we explored adjustment with maternal fat mass, which did differ between groups, and we observed similar results ( Figure S4).

  1. Results/Table 1: Please cite the original study referenced in Table 1 with the table legend. Furthermore, considering the sex-related differences in both groups (GDM- and GDM+), and the known influence of infant sex on milk composition, it would be valuable to address or at least explore this factor in the analysis or discussion.

A sentence has been added to the legend of Table 1.

In addition, we agree that considering the sex-related differences would be of great interest. However, due to the small sample size, such analyses are not possible. Much larger sample size would be needed to conduct such analyses.

  1. Figure 4 (line 324): What does the red line represent? If it indicates a significance threshold, how do metabolites such as myristic acid and uracil relate to this line?

The red line represents the expected average contribution if all variables contributed equally to the component. Variables above the line contribute more than average to the component. Metabolites like uracil and myristic acid are below this red line for Principal Component 1, meaning they contribute less to the variance explained by this dimension, compared to other variables. This does not mean they are necessarily irrelevant in the study, as these metabolites could be biologically relevant or more relevant to some other dimension or analysis. This information was clarified in the figure legend

  1. Figure 5: The variable labels are difficult to read. Please improve readability, and specify which variables refer to maternal and which to infant data.

We thank you for your comments. Changes have been made to improve the reading of the figures.

  1. Figure S1: The meaning of the p-value in the PCA analysis is unclear, especially since PCA is an unsupervised method. Please clarify.

Thank you for your comment. We do agree that PCA is an unsupervised method and that, unlike other methods, it does not provide a p-value. However, to explore potential group differences in the multivariate space defined by the PCA, we performed a MANOVA on the individual PCA scores. The p-values we report are from the MANOVA results, and they test whether the distribution of individuals in the reduced PCA space was significantly different between the groups. We used them to statistically assess whether the groups were distinguishable in PCA space.

Additions have been made to the figure legend to clarify this point.

  1. Discussion- pyroglutamate: The discussion of pyroglutamate variability could be strengthened by addressing its role in glutathione metabolism. Is there literature connecting pyroglutamate or glutathione deficiency to GDM?

As suggested, additions have been made on pyroglutamate and its role in GDM in the manuscrit discussion. 

Lines 767-773 : « The only metabolite to be significantly lower in GDM+ milk than GDM- was pyroglutamate. Pyroglutamate is a derivative of the amino acid glutamate (glutamic acid) and has been observed to be decreased in people with type 2 diabetes [4]. In the literature, decreased levels of pyroglutamate have been observed in the serum of pregnant women with GDM+ compared to GDM- control women [5]. These changes also seem to be reflected in GDM+ human milk according to our results (Figure 2). Since pyroglutamate is a component of the glutathione cycle, its decreased level in GDM+ milk could reflect a disturbed maternal metabolic state (oxidative stress). »

  1. Discussion-milk cholesterol: Several studies have suggested beneficial effects of cholesterol in breast milk compared to formula. A brief discussion of this relation to your findings would add valuable context

We thank you for your comment and have added information regarding the effects of cholesterol in breast milk compared to formula.

Lines 761-767 : “The higher cholesterol level in GDM+ human milk may be due to altered cholesterol absorption and HDL maturation in GDM+ women [6]. The altered lipid profile of GDM+ mothers could also explain the elevated cholesterol levels in their milk. Studies highlight the importance of cholesterol uptake from human milk, compared to milk formula, in regulating endogenous cholesterol in adulthood [7,8]. However, the long-term effects of high cholesterol levels in human milk on infant growth remain unclear and require further research.”

  1. Neville, M.C.; McFadden, T.B.; Forsyth, I. Hormonal Regulation of Mammary Differentiation and Milk Secretion. Journal of Mammary Gland Biology and Neoplasia 2002, 7, 49–66, doi:10.1023/A:1015770423167.
  2. A.Abdul Sattar, S.; H.Abdulla, A.; Sh. Nsaif, A. Gestational Diabetes Mellitus and Hormonal Alteration. IJPS 2017, 25, 37–41, doi:10.31351/vol25iss1pp37-41.
  3. Fradet, A.; Castonguay-Paradis, S.; Dugas, C.; Perron, J.; St-Arnaud, G.; Marc, I.; Doyen, A.; Flamand, N.; Dahhani, F.; Di Marzo, V.; et al. The Human Milk Endocannabinoidome and Neonatal Growth in Gestational Diabetes. Front. Endocrinol. 2024, 15, 1415630, doi:10.3389/fendo.2024.1415630.
  4. Zhang, Y.; Zhao, H.; Liu, B.; Shu, H.; Zhang, L.; Bao, M.; Yi, W.; Tan, Y.; Ji, X.; Zhang, C.; et al. Human Serum Metabolomic Analysis Reveals Progression for High Blood Pressure in Type 2 Diabetes Mellitus. BMJ Open Diab Res Care 2021, 9, e002337, doi:10.1136/bmjdrc-2021-002337.
  5. Kong, X.; Zhu, Q.; Dong, Y.; Li, Y.; Liu, J.; Yan, Q.; Huang, M.; Niu, Y. Analysis of Serum Fatty Acid, Amino Acid, and Organic Acid Profiles in Gestational Hypertension and Gestational Diabetes Mellitus via Targeted Metabolomics. Front. Nutr. 2022, 9, 974902, doi:10.3389/fnut.2022.974902.
  6. Zeljković, A.; Ardalić, D.; Vekić, J.; Antonić, T.; Vladimirov, S.; Rizzo, M.; Gojković, T.; Ivanišević, J.; Mihajlović, M.; Vujčić, S.; et al. Effects of Gestational Diabetes Mellitus on Cholesterol Metabolism in Women with High-Risk Pregnancies: Possible Implications for Neonatal Outcome. Metabolites 2022, 12, 959, doi:10.3390/metabo12100959.
  7. Bayley, T.M.; Alasmi, M.; Thorkelson, T.; Krug-Wispe, S.; Jones, P.J.H.; Bulani, J.L.; Tsang, R.C. Influence of Formula versus Breast Milk on Cholesterol Synthesis Rates in Four-Month-Old Infants. Pediatr Res 1998, 44, 60–67, doi:10.1203/00006450-199807000-00010.
  8. Robinson, S.; Fall, C. Infant Nutrition and Later Health: A Review of Current Evidence. Nutrients 2012, 4, 859–874, doi:10.3390/nu4080859.